# An In Vivo Proposal of Cell Computing Inspired by Membrane Computing

Alberto Arteta Albert [1,*,†,‡], Ernesto Díaz-Flores [2,‡], Luis Fernando de Mingo López [3,‡], Nuria Gómez Blas [3,‡]

1 College of Arts and Sciences, Troy University, Troy, AL 36082, USA
2 Department of Pediatrics/Oncology, University of California, San Francisco School of Medicine, San Francisco, CA 94158, USA; ernesto.diaz-flores@ucsf.edu
3 Escuela Técnica Superior de Ingeniería de Sistemas Informáticos, Universidad Politécnica de Madrid, 28031 Madrid, Spain; fernando.demingo@upm.es (L.F.d.M.L.); nuria.gomez.blas@upm.es (N.G.B.)
* Correspondence: aarteta@troy.edu
† Current address: College of Arts and Sciences, Troy University, 129-A MSCX, 600 University Avenue, Troy, AL 36082, USA.
‡ These authors contributed equally to this work.

**Abstract:** Intractable problems are challenging and not uncommon in Computer Science. The computing generation we are living in forces us to look for an alternative way of computing, as current computers are facing limitations when dealing with complex problems and bigger input data. Physics and Biology offer great alternatives to solve these problems that traditional computers cannot. Models like Quantum Computing and cell computing are emerging as possible solutions to the current problems the conventional computers are facing. This proposal describes an in vivo framework inspired by membrane computing and based on alternative computational frameworks that have been proven to be theoretically correct such as chemical reaction series. The abilities of a cell as a computational unit make this proposal a starting point in the creation of feasible potential frameworks to enhance the performance of applications in different disciplines such as Biology, BioMedicine, Computer networks, and Social Sciences, by accelerating drastically the way information is processed by conventional architectures and possibly achieving results that presently are not possible due to the limitations of the current computing paradigm. This paper introduces an in vivo solution that uses the principles of membrane computing and it can produce non-deterministic outputs.

**Keywords:** membrane computing; bioinformatics; unconventional computing

## 1. Introduction

This paper presents a new biocomputational paradigm influenced by cell computing and in particular inspired by Membrane Computing [1,2] called MECOMP.NET. We aim to exploit eukaryotic cells as processing units to model/predict and quantify in silico any of their many biological functions measurable in time and intensity using computing algorithms within MECOMP.NET. The significance of this paper is that it would represent a quantum leap in the field of biomedicine, as it would accelerate the areas of drug discovery, disease understanding, and biological process understanding.

Many cellular properties act as biological processing units as they are great conductors, communicators, and potential parallel processing units [3]. The direct parallelism between cells and computational processing units is precisely the root of the MECOMP.NET project. This paper proposal performs an evolutionary jump in comparison with today's research in this field by turning cells into computational devices. The emergent properties and functions of the cells (parallelism, no determinism, electrical conductivity, communication, and cooperation) can be used for the resolution of computationally intractable problems, not appropriately solved by conventional methods. The best conventional models that they can currently do is to obtain approximations. It is clear that for polynomially bounded problems,

the conventional approach might be optimal. However, complex and computational intractable problems would be a better fit for biological processing units [4].

The studies cited in this paper show that eukaryotic cells have already demonstrated potential processing properties. In particular, cell membranes have been proof to work as single CPU units. A system with 10 billion cells could potentially work as a system with 10 billion CPUs working together to deal with computational problems. A future implementation in laboratory settings of a computational system trained to link changes in parameters (that occur within seconds or minutes after adding an input) to resulting changes in cellular functions or outputs (that happen within hours or days) is accelerating research discoveries in an unprecedented way.

MECOMP.NET shows the potential to go beyond classical bio-computing strategies such as self-reproducing machines [5], cellular automata [6,7], multilayer perceptrons and neural networks [8,9], genetic algorithms [10,11], adaptive computing [12], bacteria-based computation [13,14], and artificial cells [15]. Interestingly, these models are not just speculative or hypothetical, the state-of-the-art in this proposal shows a large number of studies that solidly support the possibility of creating such systems. The computational properties of cells are proven, the theoretical computational models are correct, and some implementations in the lab have already been successfully tested. The proposal builds upon prior work and focuses on what is still needed to tackle high complexity problems. Specifically, building a new generation of natural computing based upon the scalable "minimal biological units" with problem-solving capacity in very different realms.

New problems are generated every day, increasing the need for processing massive data in a short amount of time. Conventional architectures have been facing the inherent limitations of the traditional framework for several years. Thus, integrating biological approaches like MECOMP.NET will be increasingly demanded to satisfy the needs or new applications in Information Systems.

This paper introduces a systematic cellular computing approach starting by establishing the principles to be computed to generate robust predictions of high complexity biological functions. Thus, the first goal is to determine such basic principles to be computed using the minimal computing units. Those principles may be intracellular events, such as protein modifications (phosphorylation), ion release from organelles (through ion channels), and protein expression, or cellular functions: proliferation, cell cycle arrest, and cell death. The second goal is to detect processes of higher complexity that can be used as circuits or networks of information. The tests in the lab will follow the design of models that are computationally and theoretically correct.

A brief description of MECOMP.NET platform, together with the fundamental functional blocks of cell processing, which are tentatively included in the present scheme are the following, see Figure 1:

1. Metabolism (MET)
2. Supermetabolism (SUP)
3. Signaling pathway system (SIG)
4. Transcriptional and epigenetic gene control (TRC)
5. Ubiquitin protein degradation (DEG)
6. Cytoskeleton and cellular adhesion (CSK)
7. Cell cycle regulation (CCR)
8. Genome profile (GEN)

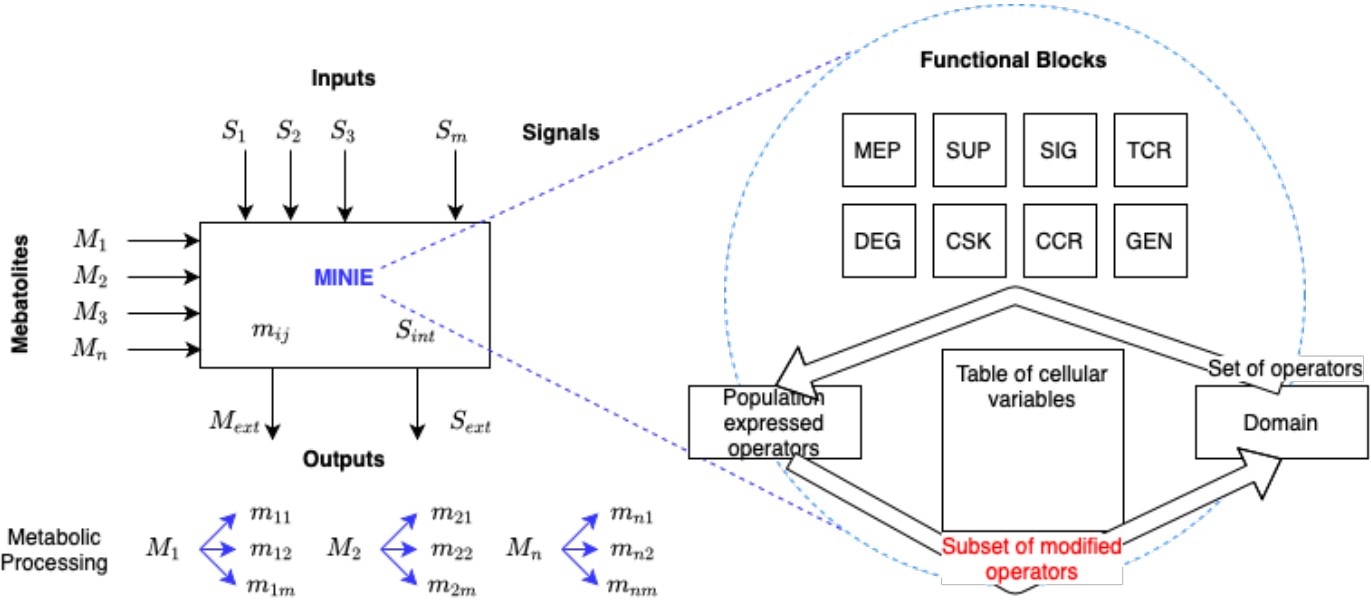

**Figure 1.** MECOMP.NET diagram: it receives metabolic ($M$) and signaling inputs ($S$), processes them ($m_1, m_{12}, \cdots, S_{int}$) in a single cell unit represented as (MINIE), generating metabolic ($M_{ext}$) and signaling outputs ($S_{ext}$).

Inside MECOMP.NET (see Figure 1) a look-up table of cellular variables (comprising all inputs, outputs, and intermediate states) is continuously updated and put into contact with the population of expressed operators, being part of some of those operators of the table itself, as they are modifiable by other operators. The interaction between the population of expressed operators and the state variables (including modifiable operators) constitutes the computational core of MECOMP.NET. This operator can be activated by some molecular components including nutrients (such as fetal bovine serum (FBS)), drugs, promoter inducers (Doxyclin), enzymatic substrates (Luciferin), or electrical/electromagnetic signals [16].

The overarching goal of this work is to provide feasible computational solutions that overcome current biological limitations to achieve higher scientific discovery rates. A current unmet need in scientific research is the prediction of biological outcomes resulting from a chain of events triggered by biosignals, nutrient processing, drug response, or electrical signals in a similar way that any of the bioinspired computational models do. Providing such a framework paves the path for building biology-inferred computers that can potentially deal with relevant questions in biomedical fields that are not intractable, costly, or time-consuming with current methodologies in a much expedited and efficient way.

## 2. Background and Prior Work

Synthetic biology has built robust models, and simulated complex circuits, using in vivo models and thus has become extraordinarily informative for the manufacture of biological components not naturally occurring (artificial chromosomes), or to scale production of natural components (DNA molecules, proteins, etc.) [17]. In the first wave of these studies, fundamental elements such as promoters, transcription factors, and repressors were combined to form small, simple modules with specified behaviors. In 1999, W.L. Ditto created a biocomputer at Georgia Tech that was capable of performing simple additions with these simple modules. Currently, biological modules include switches, cascades, pulse generators, oscillators, spatial patterns, and logic formulas [18]. In 2013, biological transistors were designed to build AND, NOT, and OR biological gates. The latter has been recently implemented with success [19]. These findings opened a new way of replicating the conventional CPU units of in vivo materials. The new ways of automatically counting cell components, such as density, number of regions, calcium molecules, and bacterial tissue [20], make it very possible to process a vast array of biological outputs and com-

bine them with conventional models, producing hybrid solutions (in silico and in vivo integrated systems).

George Păun considered membrane computing processes as basic calculator processes or basic computing units. The model opened new ways of researching when solving NP-complete problems by generating theoretical parallel processing units. Membrane Computing is commonly referred to as Transition P-system and is inspired by biological dynamics; however, Transition P-systems have always been used as a computational model, instead of a biological model. Presently, there are many simulations in silico of such systems, but unfortunately there are no attempts or approximations of membrane computing in vivo. There are, however, some implementations of in vivo computing in related fields, such as cellular computing that uses unicellular organisms called ciliates. Ciliates, for instance, store a copy of their DNA containing functional genes in the macronucleus, and another encrypted copy in the micronucleus. From the biological point of view, a plausible hypothesis about the bioware that implements the gene assembly process was proposed based on template-guided recombination [19].

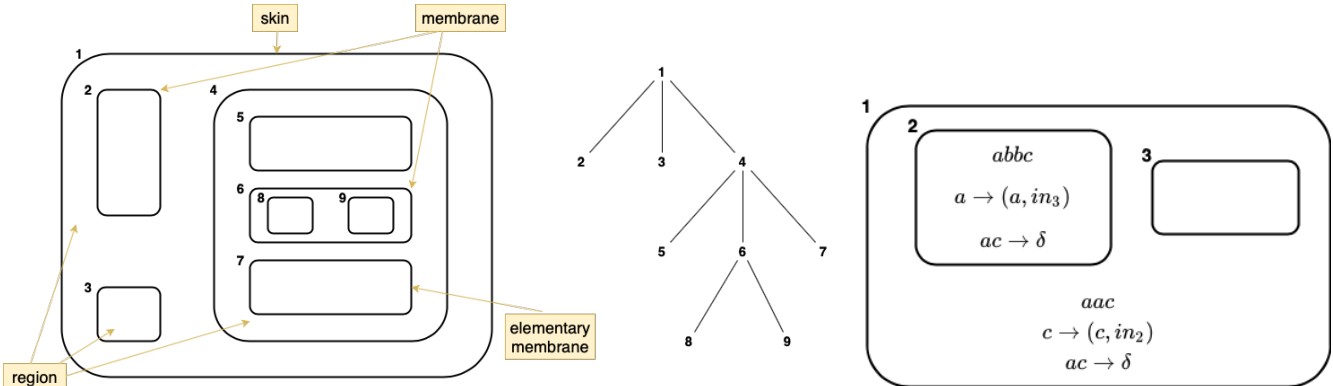

**Figure 2.** Representation of basic components in a membrane system that could be transformed into a tree shape using is-compound-of relationships, and an example with object multisets and evolution rules (from left to right).

Formally speaking, a Transition P-system of degree $n, n > 1$ is a construct Where $V$ is an alphabet; its elements are called objects; $\mu$ is a membrane structure of degree $n$, with the membranes and the regions labeled in a one-to-one manner with elements in a given set, see Figure 2; and Transition P-systems evolve accordingly to the evolution rules application in several membranes. An example of the evolution rule is $aab \rightarrow (a, here)(b, out)(c, here)(c, in)$. After using this rule in a given region of a membrane structure, two copies of $a$ and one $b$ are consumed (removed from the multiset of that region), and one copy of $a$, one of $b$, and two of $c$ are produced; the resulting copy of remains in the same region, and the same happens with one copy of $c$ (see Figure 3).

Due to the transformation that transition P-systems have undergone over the years, some reports discuss deterministic P-systems [21] and P-systems with minimal parallelism [22], and algorithms for applying evolution rules are being improved [23]. In a report by Arteta et al. [24], a computational model inspired by the aggregation of membrane units is shown to potentially work as a problem resolution solver.

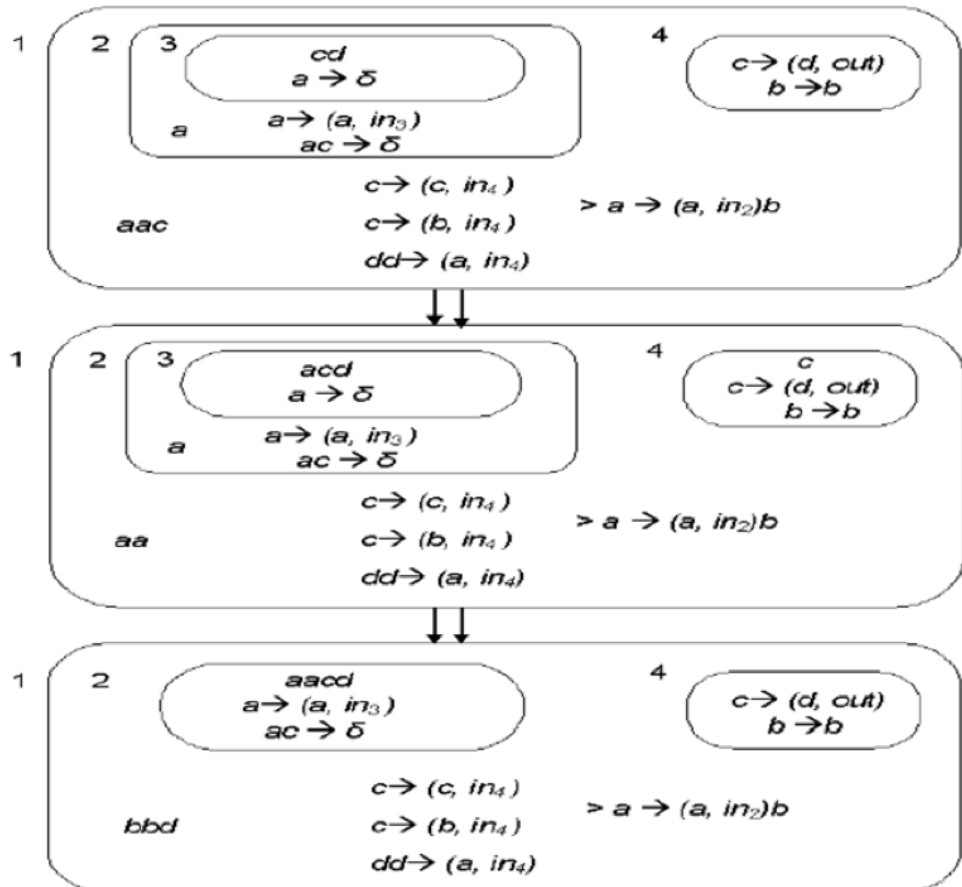

**Figure 3.** Dynamics of a P-System, also named evolution. Evolution rules in any membrane can be applied in parallel provided antecedent constraints belongs to its object multiset. The next evolution step starts when all rules have been applied in parallel.

### Ad Hoc Hardware Solutions

The implementation occurs directly on hardware tools designed especially for that purpose (usually FPGA technology). In this scenario, Păun reminds us: "The implementation of a P-system in an electronic computer is not an implementation itself but a simulation". So far, parallel hardware does not exist. Simulations lose the main advantages of the P-systems [1,2,25]. Other researchers [26] show the model and the design, but it does not go beyond a P-system's implementation. Petreska et al. [27,28] described the first implementation of hardware for FPGA. Here, it is possible to input all the parameters involved in P-systems. However, the system is deterministic, which falls short of the requirements. Thus, it is the first implementation merely in hardware, but it is not deterministic. Moreover, the hardware is only possible to be used for a particular P-system. What is needed is to use universal hardware ready to be used with any P-system. There are also some ideas about a non-deterministic proposal of universal hardware for any P-system through FPGAs [29]. The introduced hardware ensures non-determinism and universality. However, the application of evolution rules is not a massively parallel process. Nguyen et al. [30] provide a solution that looks like the one proposed in [28], with the main difference being that it is possible to implement parallelism efficiently over reconfigurable hardware.

Solutions based on ad hoc with universal hardware models need software development (low-level programming language most of the time) that will be running over hardware components. Those components are designed specifically for that purpose. They will have similar characteristics to the microcontrollers present in smart cards.

No P-system has been implemented in vivo, but some cell computing of membrane-related models has been very close. In 1994, Adleman accomplished the first experimental

close connection between molecular biology and computer science [31]. He described how a small instance of a computationally intractable problem might be solved via a massively parallel random search using molecular biology methods. A recent study [32] demonstrated that the steep transmembrane ion gradients in eukaryotes are critical for receiving and processing environmental information. Information is received when some perturbation causes the protein gates in transmembrane ion channels to open. This feature is essential to induce electronic signals into membrane units and propagate them within the eukaryotic cell and opens a solid possibility of integrating some biological models within in silico solutions.

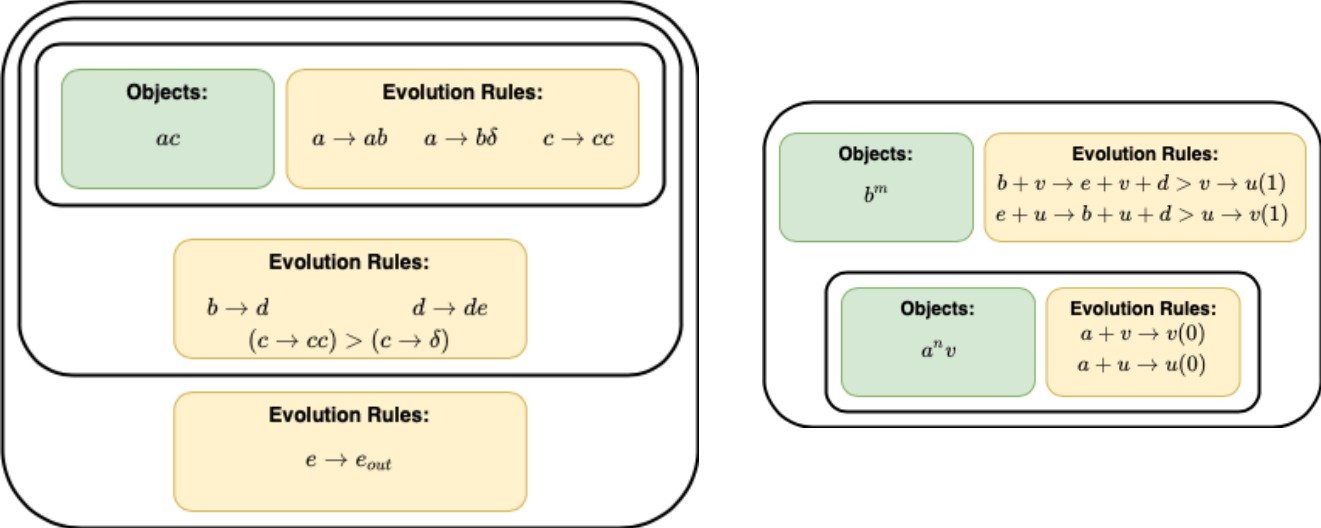

**Figure 4.** The graphical representation of a P system which outputs square numbers into the environment [25] (**left**) and multiplication [33] (**right**).

Biological P-systems will play an essential role in creating biological computers [1,2,25].

## 3. Membrane Computing: Integration in Cell Computing

Cell computing started as a framework and a simulation in silico of cell processing as a whole. This framework was proven to be theoretically correct, and the simulations have been successful as stated in the state-of-the-art. Based on in vivo computing simulations, in vivo implementations (in lab) have been successfully done. They represent a real implementation and there are several applications already in place (see state-of-the-art). Membrane computing is related to the cells computing paradigm, although very specific. This model was proven by George Păun as a theoretically correct computational model, and there are numerous simulations as previously stated. However, there is no implementation of this specific paradigm in a laboratory setting. The chances to create massive non-deterministic in vivo models of parallel processors inspired in membrane computing are relatively high, as there are successful implementations of closed related models such as cell computing. A prior study demonstrated [32] that cells have proven to be able to integrate the communication between biological units and in silico-based devices, which opens the door to create hybrid (biological in silico) computers and boost the chances of the overall success of MECOMP.NET.

## 4. Problem-Solving Characteristics

This artificial chemistry and the signaling and scalability procedures, along with the characteristics of the different membrane processing-inspired software is to be of great relevance. MECOMP.NET incorporates the emerging functions covered by networks of membrane models with a more sophisticated aggregation of membranes (MCA) [24]. From a long-term perspective, the most suited problems to be initially addressed by the

new paradigm are those of combinatorial optimization. These are usually present in Numerical Analysis, Deep Learning algorithms, Massive Data applications, and intractable computational problems, as the well-known knapsack problem, which arises in different fields: Combinatory, Complexity Theory, Cryptography, and Applied Mathematics. The problem is known to be NP-complete which means no algorithm can be both correct and fast (polynomial-time) for all cases. Besides, there are several applications of cell computing [23,34]. These are applications of hybrid simulations of cell computing successfully proving the correctness of the method. Besides, there is other related research on effective analysis of Bioinformatics data in different formats [35–38] and big data [39,40]. The evaluation of the complexity and universality characteristics of MECOMP.NET as a new computational paradigm implies examining computational relationships and convergences of MECOMP.NET with cellular automata, evolutionary agents, and the Turing machine. These systems are evaluated in terms of algorithmic convergence as a transformation of emergent properties of the proposed architecture.

## 5. Methodology

Our proposal can be divided into four main methodological blocks which contain phases that at times will be developed in parallel: biological analysis (phases 2, 4, and 5), computational simulations (phases 1, 3, and partly 4), and evaluation plan (phase 6).

### 5.1. Phase 1

A theoretical study, modeling, and formalization of membrane computing (initial simulation): Any biological network, either at the "cellular" or "tissue" level shows an evolvable development and differentiation, and that is what will be simulated in this phase. In this phase, simulations in ISLISP and Haskell 7.6 (Haskell.org Inc., P.O. Box 1206, New York, NY 10159-1206, USA) are running to identify the different states the system will evolve with the membrane computing principles; this helps to design the in vivo system algorithms. This design is essential, as it establishes the basis of creating a computational paradigm in the lab.

### 5.2. Phase 2

Bioinformatics implementation of simulations to the biomolecular processes in vivo: The standard model is not fully replicable in the lab. Adapting the theoretical framework according to the biological rules that the cell regions are limited is the goal of this phase. An exhaustive study of standard rules, finding the feasible biological inputs and outputs, is needed to apply it and to adapt it to the lab requirements. An abstraction of these basic properties is set up the guidelines for the creation of a minimal set of cellular units. Cellular components are considered either active or passive and we will use production rules (molecular, cellular, and tissue operators, enzymatic and self-assembly operators, and so on) to measure the interaction between these components and the outcomes with some similarities to the way P-systems evolve.

### 5.3. Phase 3 (Laboratory Experimentation)

Implementation of biological rules (input, evolution process, output, and execution time): The knowledge needed for the correct advancement of this phase is extracted from a deep analysis of the components and interactions that take place in cell membrane units found in phase 2. Based on the evolving patterns found in the previous phase, this stage focuses on the massive testing of those rules that include the previously detected evolving patterns. The goal of this phase is to find and recording information vectors (input, evolution rules, output, and computation time). At the end of this phase, the experimenter reports a large number of computation rules that respond to the evolving patterns found in phase 2. The experimenter will focuses on the rules that can be fully replicated from the transition P-Systems.

Examples of potential programmable rules to test in the lab are below. These rules are defined in a similar way to the evolutionary rules within the Membrane Computing computational paradigm. These have been tested in the lab. Initially, the biological rules extracted from a membrane system in Figure 4 are simulated to obtain square numbers and basic multiplications. The dosage of FBS (or the amount of EE electrical signals) is measured to obtain a similar behavior of the evolution rules in Figures 5 and 6. The application of these reactions helps to define the proper evolution rules in biological systems.

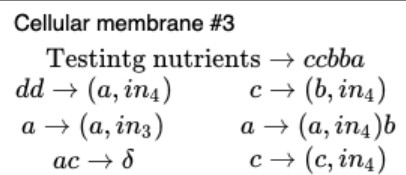

**Figure 5.** A three-membrane system brings a simple membrane system designed to calculate a random number.

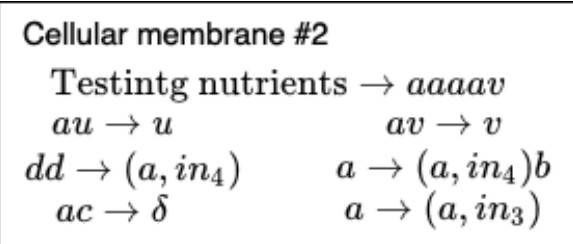

**Figure 6.** Two-membrane systems designed to multiply the number of b's and a's.

Initially, the tests consider application rules and multisets of objects from a membrane computing framework. The first in vivo attempt will simulate in the lab the transition P-system that returns a random value. See below examples of reactions for the initial study. The multisets elements *a*, *b* are defined to represent nutrients. The elements to be tested for these objects (*ab*, *c*) are organic, $Na^+$, $H^+$, O, and Ca, and inorganic, Fe, Se, and Zn, these are represented as "nutrients".

*5.4. Phase 4*

Synthesis of biomolecular/biocomputational interconnections of the recorded processes: Once all the rules have been tested, we prioritize those with structural similarity to the membrane computing model; some of the rules above are filtered out to consider the development of complex problems solving systems. Then, we train the system into different contexts or different descriptive levels and apply it to complex problems to establish a more advanced approach allowing the analyses of problems at different stages, from the least concrete to the most practical. These rules are classified into feasible for membrane rule, computational feasible, non-feasible. This phase is essential for the development of the project as determines the performance and the ability to build programmable cellular units. Based on the dosage of FBS, the intracellular signaling and the principles of bioinspired computing techniques (the biological rules) dictate the evolution of the membrane units. The system then reads output signals (by quantifying elements within the membrane regions) and links them to input signals. The evolution rules are reprogrammed based on the way the cells consume the nutrients; they are simulated by adding stimuli previously defined and chosen in phases 1 and 2 and observing in what quantity this affects the input objects. This step is essential to induce the right amount of signals that determine the evolution rule.

*5.5. Phase 5 (Laboratory Experimentation)*

Design of the initial biological membrane-based minimal unit: Based on phase 3, the design of the processing is relatively complex. Programmable rules are the seed of the evolutionary development of a P-system in vivo. Those rules along with the stimuli and nutrients, working as objects, are part of the unit. As the changes of the different regions (part of phase 2) are recorded in detail, when different stimuli are given, additional features can be embedded in the unit further on. Thus, this phase also opens new possibilities to future adding to the membrane system in vivo. For this purpose, we use abstractions of active components (membrane, proteins, enzymes, etc.). These components will be gathered in the functional design to be implemented. Configuration's process in this context is based on performing a search of components previously included in a catalog and complying with the initial conditions.

*5.6. Phase 6*

Testing of the unit with rule elements of the cellular functions selected during the design process: This phase considers the results of implementing the structure (regions/functions) nutrients/objects and rules that have been found in phase 3. Every potential is tested a minimum number of ten times to offer the same results, although the non-determinism feature embedded in the P-system will also be tested in vivo. From the theoretical point of view, defining structures that offer the expected output regardless of the evolution rules are also explored as membrane computing by itself is a limited framework and a full replica in vivo is practically impossible [41]. The development of the computation paradigms for high-performance distributed computing as well as the need for increased computational power to solve complex problems is the key that inspires this phase. The tests are done to bring resolution to simple problems such as multiplications by adding two inputs.

**6. Evaluation and Results**

The proposal has been evaluated based on the following criteria: Evaluation of the selected rules: evaluation of the system is done in terms of algorithmic convergence as a function of emergent properties of the proposed architecture. A crucial related aspect will be a refinement of the software that will support emerging processes of multicellular systems. The in-lab synthesis of ad hoc "minimal biological units" is studied as a different proof of theoretical designs and software simulations. The tests are done with trial-and-error attempts and will consist of three main phases.

Rule detection for a processing unit: Rules include programming the cells, observing the changes in different levels (components, objects proliferation, density, and many other changeable attributes within eukaryotic cells), timing the process, and reporting the results. Timing, parallel degree of transformations, and quantification of transformations are crucial, and we will relate them successfully with three dosages of the applicable drugs: inducers.

During the evaluation, the rules defined in phase 3 have been tested to detect the possible biological processing units. The rules that do not contribute to anything programmable are discarded. The rules that simulate evolutionary rules in computational models are tested first.

A rule has been considered to be part of a biological processing unit if and only if the following hold.

- Timing of returning output is acceptable in comparison with traditional computing paradigms.
- The process occurring in the cell transformation that produces the output in parallel.
- It is possible to find the relation between the component changes and the intensity/amount/number of stimuli used (drugs, electrical signals, or inducers) to stimulate the unit.

Evaluation of the unit: Once the rules are identified and tested, the integrative model is created and the concept of the minimal processing unit is generated. The evaluation consists of repeating the procedure input/outputs) according to a looping parameter ($\mu$),

initially defined as μ = 10,000. This variable determines the number of times (initially 10,000) an input transformed into an output according to the selected evolution rules. The goal is to identify the deterministic degree of the models and to verify the same rules, with the same stimuli (dosage of FBS, inducers, or EE signals) and same inputs produce expected outputs, and therefore the test leaves the building process of the processing units. The potential units we have initially considered as biological CPU units are not healthy cells but leukemic cells, mainly due to their proliferation rate that can boost the simulation performances. The three B cell leukemia cell lines that are used during this study are NALM-16, Beck-1732, and MHH-CALL2 (The NALM-16, Beck-1732, and MHH-CALL2 cell lines were obtained from Dr. Ernesto Díaz-Flores [42]). All three leukemia cell lines belong to the subgroup of hypodiploid leukemia. Their duplication time is about 24 h when grown in RPMI culture medium in the presence of exogenous L-Glutamine and 10% Fetal Bovine Serum (FBS) at 37 degrees Celsius and 5% $CO_2$. These cells were used in a recent publication studying their genome and protein profiles, proliferation rates, and response to multiple drugs as faithful models of hypodiploid leukemia [42].

As can be observed in Figure 7, only the ABT-263 drug (a Bcl-2 inhibitor, orange) reduced viability to a large extent with concomitant induction of cell death. From a computational standpoint, those graphs indicate how a value representing the proliferation status of cells at 24 h modulated via a stimulus (drug) that operates as an add/subtract function to the protein levels are easily computed.

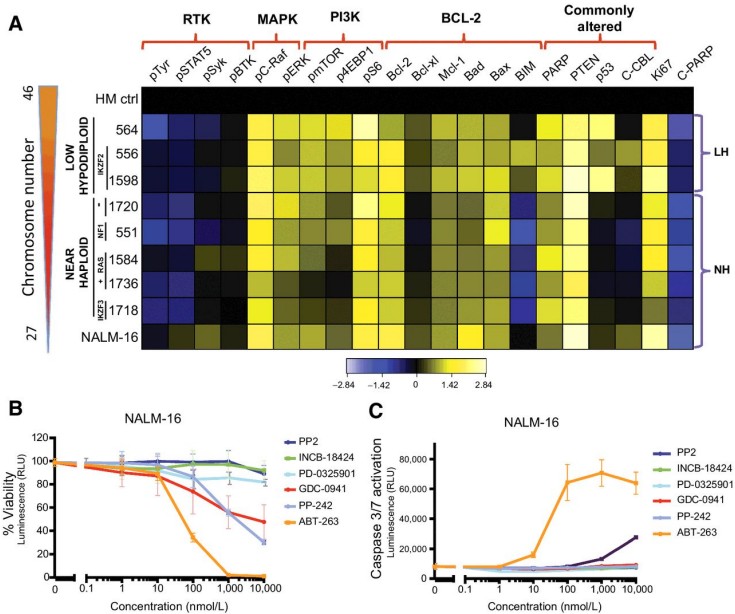

**Figure 7.** Integrated biochemical characterization with viability/survival assays in hypodiploid ALL. (**A**) Cytobank heatmap representing a signaling profiling panel of phospho- and total protein levels (columns) in individually xenografted leukemias from LH and NH subtypes (rows). Levels of phospho- and total proteins were normalized to healthy donor control cells (HM ctrl). Yellow, increased levels; blue, decreased levels. (**B**,**C**) NALM-16 cells were cultured for 24 h in the presence of increasing concentrations of inhibitors (PP2, INCB-18424, GDC-0941, PP-242, PD-0325901, and ABT-263) and effects were measured using cell proliferation (**B**) and apoptosis levels (**C**) using ATP-luminescence (CellTiter Glo) or caspase-3/7 activation assays, respectively [42] The main counters for the computational study are displayed in (**B**) cell proliferation and (**C**) ATP-Luminescence. These counters are selected for the implementation of the membrane system, as the variance in the values make them a good fit for evolving during the evolution process of the membrane system. Reproduced with permission from Ernesto Díaz-Flores, Cancer Research, published by American Association for Cancer Research, 2019 [42].

### 6.1. Results Turning Cells into Computational Units by Training MECOMP.NET with Biological Processing Inputs and Outcomes According to P Systems Evolution Rules

This stage deals with measuring cellular proliferation based on rapid changes in fluorescence. Cells are continuously processing information by transducing external stimuli into a chain of intracellular signaling. That produces a response that encompasses multiple processes: protein synthesis, cell proliferation, cell death, etc. Cells can be engineered to harbor reporter fluorescent tags that have been induced upon stimulation, allowing to tracking of multiple cellular and biological processes. Immortalized cells are used in research laboratory settings to study multiple processes from understanding cancer cell behavior to drug responses, stimuli responses, and so on. The leukemic cells to be used in this study have been tagged to express either fluorescence tags or luminescence tags to track a series of biological processes. These immortalized tagged leukemic cells are mini-processors responding to a series of stimuli.

Experimental procedure: For the work proposed here, we will use Nalm-16, an immortalized B cell leukemia, (1,2), as well as two other B-ALL cell lines (Beck-1732 and MHH-CALL2) as CPUs.

- Experimental set up (A)
  - Serum induced proliferation
  - Input: nutrients (Fetal bovine serum (FBS))
  - Output: proliferation (reporters: Ki67, ATP production)
  - Readout method: Flow cytometry analysis (Ki-67) (fluorescence), Cell titer glo (ATP) (luminescence)
  - Methodology: Cells will be arrested from proliferation by depriving them of serum (FBS) for 2 h. Cells will be seeded at 1 million per experimental well in a 6-well plate, and FBS will be added in a dose–response manner, from 0.01% to 10%. Percentage of ATP (linearly correlated to proliferation) will be recorded every 30 min for 6 h and then at 12, 24, 48, and 72 h.
  - Determine how early events (% ATP, Ki67) determine the cell proliferation and select the most accurate readout method. Evolution rule $A \rightarrow A^n$, where $n$ is a non-deterministic variable, $A$ is the cell count, and $m$ evolution rule determined by ATP and FBS dosage. The P-system for this experiment is depicted in Figure 8.

$$A^m$$

$$A \rightarrow A^n$$

$$A^n \rightarrow out$$

**Figure 8.** Membrane system with a Skin membrane, two evolution rules, and a multiset of objects. In every evolution step, the multiset of objects either expands or outputs an increased number of objects.

The proliferation rate determines the growth the of the cells population non-deterministically defined by FBS. When applying the same amount of FBS, the cell population increases. Using the Membrane Computing terminology, we will say that when the evolution rule 1 is applied the multiset of objects gets always larger, with a variation in

the increase size. In the experiment, it is noticeable that the cells count grows, but not with a constant growing rate. This rules out the design of the multiplying P-system. However, the way that the cell number increases is very similar to the accumulators in loops, where in every round the accumulator gets updated with a new value (a new value is added, but not necessarily the same one).

The experiment provides with a method for simple addition; however, the inherent non-deterministic character of the system makes it doing it in a different way a basic calculator does.

- Experimental set up (B)
    - Drug-induced cell death
    - Input: Bcl-2 (survival) inhibitor (ABT-199)
    - Output: cell death (reporters: Caspase-3)
    - Readout method: Active Caspase by Caspase Glo assay (Promega)
    - Methodology: Cells will be seeded at 1 million per experimental well in a 6-well plate. ABT-199 will be added in a dose–response manner from 0.01 μM to 10 μM. Caspase levels will be recorded every 30 min for 6 h and then at 12, 24, 48, and 72 h. Similarly to the previous experiment, the design of the Membrane system is identical with a replacement of the main evolution rule.
    - Determine how time and intensity of Caspase induction at early time points predicts long-term cell death induction. Evolution Rule $A^n \rightarrow A$, where $n$ is a non-deterministic variable, $A$ is the cell count, and $m$ evolution rule determined by ABT-99 and Promega.
    The experiment provides with a method for simple subtraction; however, the inherent non-deterministic character of the system makes it doing it in a different way a basic calculator does.

### 6.2. Experimental Setup: Relay System

This experimental module uses cells that have been engineered by the experimenter to express an inducible CRISPRi system [43] (Creative Biogene, 45-1 Ramsey Road, Shirley, NY 11967, USA). The CRISPRi system allows blocking the expression of any gene inside a cell. The researcher has access to a library of over 12,000 genes to choose from. The expression of an essential gene (Bcl-2) will be blocked. Cells were first engineered to express the inducible CRISPRi cassette. This cassette has a doxycycline-inducible Tet expression, a red fluorescent (mCherry) tag, and a ribonucleoprotein complex (dCas-9-KRAB). After adding exogenous doxycycline, both the effector dCas9-KRAB and the tag are expressed. This dCas9 effector is responsible for blocking any gene of interest. As a safeguard, and to prevent the risk of accidentally blocking the expression of any non-intended gene in the cell, this system requires the presence of another gene-specific construct (guide RNA) that will guide dCas9-KRAB to the locus of the gene of interest. The experimenter has engineered cells to express both constructs being the guide-RNA-specific for the survival gene Bcl-2. Thus, it requires the addition of exogenous Doxycycline to express dCas9-KRAB that, through the guide provided by the Bcl-2 guide RNA, will go to the genomic locus of the Bcl-2 gene and selectively inactivate the expression of Bcl-2.

The Bcl-2 gene is essential for the survival of the three cell lines above mentioned. Without Bcl-2, the process of apoptosis (programmed cell death) will be started in the cell at around 8 h, resulting in the death of the cells between 12 and 24 h. However, within the first hour upon inactivation of the Bcl-2 gene, the apoptosis machinery gets started with the rapid activation of Caspase 3. Caspase-3 activation can be detected with a great level of sensitivity using a Caspase-Glo luminescence assay.

Methodology:

- Input: 0.3 mg Doxycycline
- Output 1: mCherry fluorescence (from dCas9-KRAB expression) within seconds to minutes; output 2: luminescence from Caspase 3 within minutes.
- Plasmid information

- pHR-TRE3G-KRAB-dCas9-p2a-mCherry. This construct provide an inducible CRISPRi cassette (see https://www.addgene.org/60954/ (accessed on 8 May 2012).).
- PU6-sgRNA with EF1alpha Puro-T2A-BFP. This construct provides the target locus (see https://www.addgene.org/60955/ (accessed on 8 May 2012).) for the product of the prior plasmid.

- Goal: We will be able to determine the intensity of mCherry signal, how it amplifies over time, and the associated luminescence intensity of active Caspase. Both measures will be recorded using a dual fluorescence/luminescence Tecan plate reader. The intensity is constantly measured and it varies according to a random normalized distribution. Below there is a sample of a sequence of signal intensities captured as output and normalized by the MinMaxScaler function, see Table 1.

**Table 1.** Sample of a sequence of signal intensities captured as output and normalized by the MinMaxScaler function.

| Obtained Value |
| --- |
| 0.009173482905401653 |
| 0.08795852394045727 |
| 0.22070324396749302768 |
| 0.30095643732336056715 |
| 0.1369705775243437201 |
| 0.06725518234982978949 |
| 0.0013510424223434343248291 |
| 0.40945947152720324285 |
| 0.98775276745812343241 |
| 0.2106503392442342326 |
| 0.50411794275372344296 |
| 0.71412854805234443806 |
| 0.3233625507898234423439 |
| 0.77225188319845343433346 |
| 0.8947120410823423572 |
| 0.982603883598234428 |
| 0.34438307832664154605 |
| 0.34828934532349023208 |
| 0.6542392344384726058 |
| 0.9324078080234020215 |
| 0.9598512367857497 |
| 0.2231982390023497486 |
| 0.682921740234320202 |
| 0.8502694852342394734 |
| 0.89324584922344746 |
| 0.3552141814234356327 |
| 0.672017102940323422309 |
| 0.432234234387391175223 |
| 0.6314661023432474233462 |
| 0.108053954768242343831 |
| 0.9015822342306587772993 |
| 0.6871116663243946124 |
| 0.8664159148262345153 |
| 0.04020811408726243432 |
| 0.7907823452440830576 |
| 0.9830167234232302717 |
| 0.24379081474323423136 |
| 0.24983990528142353754 |
| 0.45571435994241243116 |
| 0.00211751432589616731 |
| 0.040086586894410026 |
| 0.659815686044232144 |
| 0.368768656599938237 |
| 0.588077512946463449 |
| 0.07206175658728346324 |
| 0.53548662860923895551 |

The sequence has been tested with the Monobit Frequency test, providing evidence of a low correlation or lack of patterns in the generation of the numbers. The membrane system design corresponds to the P-system calculating a random output include in [1,2].

The relevance of generating a computational model of CRISPR has great implications not only from the computational standpoint, but also from the biological standpoint. CRISPR has seen the fastest implementation in research laboratories worldwide. Providing researchers with a mathematical model that could be used to predict or quantify genome editing using CRISPR in any system and with any gene would be of utmost relevance and may be subjected to rapid and wide applicability.

## 7. Conclusions

This paper has been nested within a multidisciplinary effort, understood as the set of activities, services, and programs that are meant to support and provide solutions that will help society in specific areas such as Biocomputing, Computational Biology, and/or Unconventional Computing. The work done in the lab has shown that it is possible to obtain random outputs and very simple additions for biological units that use basic powered evolution rules. The use of leukemic cells also offers an advantage in the growth and creation of biological units due to their high proliferation rate. The trials have proven that nondeterministic biological processing units are possible and basic operations can be performed in an alternative way to traditional computers. The results open a door for the creation of more complex units.

Full implementation of a complex MECOMP.NET with a massive amount of nondeterministic evolution rules for different biological inputs would provide contributions in the five following areas: parallel computing, new computational paradigms, complex problem resolution, applications in bioinformatics, and cybersecurity applications.

The paper has focused on the development of new types of biocomputational systems, predictive tools for exploring combinations of differentiation signals, development of hybrid systems, theoretical developments in computer science and technology from software development to ad hoc hardware, and new approaches to problem-solving, especially regarding combinatorial optimization problems. A common theme envisioned for MECOMP.NET for a multidisciplinary impact will be to provide a new way of understanding the relationship between biology and computer science, creating hybrid systems first and possibly full biologically-driven devices later, depending on how the biological devices respond to the integration. In light of this, one of the most important long-lasting outcomes of MECOMP.NET would be the development of a new category of dynamic, interactive modeling systems, which can be used as an integrative tool for understanding, discussing, and helping to manage complex computational problems. Potential implications that are obvious and relatively easy to achieve after completion of a complex version of MECOMP.NET include the following.

A real random number generator: The conventional random generator used in simulations, data analytics, and even in some encrypted development is based on RAND libraries that are inherently deterministic and based on seed numbers manipulated by complex functions (ICG or LCG) to obtain pseudorandom numbers. The simulation in P-System architectures has shown that it is possible to produce a random number based on the random selection of evolution rules. This behavior is a replica of some of the random outputs a cell can produce based on the same stimuli (input). In summary, accomplishing this goal will have an impact in fields like cybersecurity, as it will eliminate the possibility of predicting the next random value in a sequence as a part of the encrypting key, and Data Science (offering a more reliable distribution of random populations). Simple P-systems are defined to do this. Our experiment has shown the existence of random outputs (quantifiable luminescence flashes) based on given inputs (fixed amount of Doxycycline).

A minimal processing biological unit: Without considering the performance of a membrane-based biological unit, building a minimal one is achievable and can be a new way of processing information. The massively parallel character of the transition P-systems,

brought to the lab, will theoretically offer a considerable reduction in terms of time when the input $n$ (objects/nutrients) increases, producing outputs in polynomial time $N^\alpha$ for some $\alpha \geq 1$. Each unit is represented as a cellular region with sub-regions/membranes and it is independent and autonomous units that can process low-level operations such as arithmetic, multiplications, or random. The trials presented here have been able to prove that a minimal processing unit can be constructed in the lab that performs create random numbers and performs simple arithmetic operations as in the variance of the intensity of the fluorescent processes or the cells count within the system. Further work will be needed to adjust more models to the known P-systems and to boost the performance, as the complexity for these basic operations in traditional computers is lower. The main advantage of the unit vs. the traditional computers is that the generation of randomness is more accurate, as the conventional ones are only able to generate pseudorandom outputs.

Hybrid unit: The next step for this project will be adding biological counters such as flow cytometers or image analyzers that can potentially get the outputs of the biological unit, process/digitize them, and send the signals to conventional CPUs. This will undoubtedly have an impact on society. A long-lasting effect of MECOMP.NET success is to open a particularly innovative direction that during the last thirty years has not been sufficiently developed, despite the highly qualified research performed by several scarcely connected groups with expertise in Natural Computing such as Membrane Computing.

In summary, this study have been able to demonstrate a basic biological unit that with a single amount of evolution rules inspired in transition P-systems, have been able to produce random outputs and small additions, which in the long term could generate a revolution in areas that require massive data processing in real-time. Regardless of the performance and feasibility of a more complex and efficient design, this can be a good keystone for the creation of more advanced biological units inspired by membrane computing.

**Author Contributions:** Conceptualization, A.A.A. and E.D.-F.; methodology, N.G.B.; software, L.F.d.M.L.; validation, E.D.-F., A.A.A. and N.G.B.; formal analysis, L.F.d.M.L.; investigation, L.F.d.M.L.; writing—original draft preparation, A.A.A. and E.D.-F.; writing—review and editing, N.G.B.; supervision, L.F.d.M.L. All authors have read and agreed to the published version of the manuscript.

**Funding:** This research received no external funding.

**Institutional Review Board Statement:** Not applicable.

**Informed Consent Statement:** Not applicable.

**Data Availability Statement:** No new data were created or analyzed in this study. Data sharing is not applicable to this article.

**Conflicts of Interest:** The authors declare no conflict of interest.

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
