# Peer review of "An In Vivo Proposal of Cell Computing Inspired by Membrane Computing"

_processes, doi:10.3390/pr9030511_

Round 1
Reviewer 1 Report
Review of the article: "An in-vivo proposal of cell computing inspired by membrane computing: MECOMP.NET".
This multidisciplinary article is on the border of science, art and the future vision of science and its use. The article is carefully and clearly written.
In summary, this is very stimulating article and in my opinion it can be published in Processes after small correction of typographical mistakes.
Author Response
Dear reviewer,
Thank you so much for your feedback. Please find attached the response to your comments

Reviewer 2 Report
The authors layout a proposal to build/design biological computational units. They claim to have created a biological computational unit that can allegedly produce random numbers. There is either little data to support this or no explanation to guide the reader as to how these values can be supported as “random”. Using the context provided, one could have used data from any study and claimed random numbers.
Line 458. Please provide additional context and constraints to support the claim that the computational unit generates random numbers. If the values are “random” to 2 significant digits, what differentiates error from random numbers?
Line 471. Please provide additional context or description as to what experimental outcome supports simple addition.
Some general suggestions would be: Make sure the figure is connected to the text. The only data figure provided is presented in a way that does not indicate how it could be interpreted to support the concepts being demonstrated (a.k.a., claims). Only two pieces of data in the data figure are mentioned in the text (and only passively) and there is no information to help the reader understand what those data mean with respect to the claims being made.
Author Response

(The authors gave the same response as above.)

Round 2
Reviewer 2 Report
Perhaps a biological computer does not follow basic calculator rules, but the proposed addition/subtraction claim is not clear in the paper. How does the change, which is said to be "protein levels" (line 351), "proliferation rate" (line 350 and 382), and "cell populations" (line 383), indicate addition/subtraction as compared to multiplication/division or even nonlinear changes? The addition/subtraction operation has a specific accepted definition that should be proven to be claimed.
Thank you for clarifying how the numbers were tested and defined as random. Please provide information about how this feature was engineered to be random. In lines 423-440, the authors describe a fairly standard reporter system and then claim the output to be random numbers as supported by the monobit test. It is not obvious how this system was designed to be random any more than biological data varies.
